# Microfluidic Device for Droplet Pairing by Combining Droplet Railing and Floating Trap Arrays

**DOI:** 10.3390/mi12091076

**Published:** 2021-09-06

**Authors:** Margaux Duchamp, Marion Arnaud, Sara Bobisse, George Coukos, Alexandre Harari, Philippe Renaud

**Affiliations:** 1Laboratory of Microsystems LMIS4, Ecole Polytechnique Fédérale de Lausanne (EPFL), CH-1011 Lausanne, Switzerland; margaux.duchamp@epfl.ch; 2Department of Oncology, Ludwig Institute for Cancer Research, Lausanne University Hospital, University of Lausanne, CH-1066 Lausanne, Switzerland; marion.arnaud@unil.ch (M.A.); sara.bobisse@unil.ch (S.B.); george.coukos@chuv.ch (G.C.); Alexandre.Harari@chuv.ch (A.H.)

**Keywords:** droplet, microfluidics, pairing

## Abstract

Droplet microfluidics are characterized by the generation and manipulation of discrete volumes of solutions, generated with the use of immiscible phases. Those droplets can then be controlled, transported, analyzed or their content modified. In this wide droplet microfluidic toolbox, no means are available to generate, in a controlled manner, droplets co-encapsulating to aqueous phases. Indeed, current methods rely on random co-encapsulation of two aqueous phases during droplet generation or the merging of two random droplets containing different aqueous phases. In this study, we present a novel droplet microfluidic device to reliably and efficiently co-encapsulate two different aqueous phases in micro-droplets. In order to achieve this, we combined existing droplet microfluidic modules in a novel way. The different aqueous phases are individually encapsulated in droplets of different sizes. Those droplet populations are then filtered in order to position each droplet type towards its adequate trapping compartment in traps of a floating trap array. Single droplets, each containing a different aqueous phase, are thus paired and then merged. This pairing at high efficiency is achieved thanks to a unique combination of floating trap arrays, a droplet railing system and a droplet size-based filtering mechanism. The microfluidic chip design presented here provides a filtering threshold with droplets larger than 35 μm (big droplets) being deviated to the lower rail while droplets smaller than 20 μm (small droplets) remain on the upper rail. The effects of the rail height and the distance between the two (upper and lower) rails were investigated. The optimal trap dimensions provide a trapping efficiency of 100% for small and big droplets with a limited double trapping (both compartments of the traps filled with the same droplet type) of 5%. The use of electrocoalescence enables the generation of a droplet while co-encapsulating two aqueous phases. Using the presented microfluidic device libraries of 300 droplets, dual aqueous content can be generated in less than 30 min.

## 1. Introduction

Droplet-based microfluidics is a sub-branch of microfluidics, characterized by the generation and manipulation of discrete volumes of solutions, generated with the use of immiscible phases. Each droplet is a microreactor unit that can be independently controlled, transported and analyzed. Thus, parallel and fast processing of multiple identical microreactors can be performed, which is of interest for various biological and chemical applications. Indeed, droplet microfluidics provide efficient volume control, fast reactions and high throughput at miniaturized scale. Of the multiple technological elements, high control over the droplet content is required.

In this study, we developed a novel microfluidic chip that can reliably and robustly co-encapsulate two aqueous suspensions in a single droplet without any sample loss, by combining five different droplet manipulation elements [1] (droplet generation, trafficking, filtering by size, pairing and merging). Additionally, the microfluidic device presented enables a retrieval of the dual content droplets generated and also provides versatility to the user by enabling the addition of other droplet manipulation elements.

Size-based filtering of droplets is an important step in the isolation and identification of different content droplets. One method to sort droplets based on their size consists of passive migration using lateral flows in different directions to the main flow [2]. In order to further improve the sorting efficiency and reliability, recent studies also added structural features [3,4,5] besides the lateral flow effects. Another droplet size selective method relies on physical filtration features [6] where structural features prevent the flow of larger droplets. However, the most widely spread droplet size-based sorting combines two passive mechanisms: Laplace’s trapping mechanism and the hydrodynamic drag force. This method implies the use of structured features located at the top or bottom of the microchannels to guide the confined droplets through the use of continuous [7] or “dotted” guiding tracks [8,9]. Yoon et al. [7] developed the “railing” method for size-based filtration with the aim of simplifying droplet sorting, which was previously performed using optical control of droplet content and active control sample systems (based on electric, ultrasonic, hydrodynamic or optical forces). In a later study, Yoon’s team also suggested the used of dotted lines instead of rails [9]. The railing system was recently further developed and complexified by Rehman [8], using multiple rails to discriminate more than two droplet sizes.

Other important microfluidic droplet elements are the pairing and merging of two droplets. While the pairing mostly relies on passive mechanisms, droplet fusion is often performed using an active mechanism. On the one hand, droplet pairing can be performed in flow where droplets of different types are placed alternatively one after the other [10,11]. On the other hand, droplet pairing can also be perform statically in an array containing trapping [12] features where droplets are confined [13,14] or just anchored [15,16]. The generated droplet pairs can actively be merged using an electric field [17], a magnetic field [18] and chemicals [19], or passively joined through channel geometrical changes [20].

The novel microfluidic device presented here combines the various microfluidic elements cited above, combining passive and active droplet manipulation fluidic modules (Figure 1). The co-encapsulation is achieved by generating two water-in-oil droplet populations of different sizes. One aqueous phase will be enclosed in big droplets while the second suspension will be placed in smaller droplets. Each droplet population is then filtered in order to position the droplets along their respective trapping sites. Pairs of droplets composed of a single droplet from each size are generated. Those pairs are then merged, which generates a library of droplets co-encapsulating two aqueous phases. Those droplets could then be removed from the traps and further analyzed. To our knowledge, this is the first device that efficiently combines a droplet railing system and a floating trap array for improved droplet pairing and merging with the aim of generating dual content droplets. The work provided in this study opens new horizons for dual content droplet generation, which could be of prime interest when working with particles (cells, beads). Indeed, having a precise control over the content of the droplet is critical when investigating cell–cell or cell–bead interactions.

## 2. Materials and Methods

### 2.1. Working Principle

#### 2.1.1. Droplet Generation

We first sequentially encapsulated two aqueous phases into two different droplet types (distinguished by their size). Our droplet generation system (Figure 2i) is based on a quasi 2D design with two 90° side channels implementing the flow focusing principle. This passive droplet generation system is extensively used thanks to its stability and reproducibility in terms of droplet size [21]. The aqueous phase channel and the droplet channel are 35 μm wide while the oil inlets are 50 μm wide.

#### 2.1.2. Droplet Railing

Once the droplets are generated, they are directed towards the trapping and pairing area. Our goal is to generate pairs of droplets, each one containing a different aqueous phase. In order to prevent any droplet loss (in low flow areas), on their way to the trapping area we decided to guide the droplets using rails. By creating etches (40 μm deep and 10 μm wide) in the microchannel top wall, called rails, it is possible to guide droplets along the resulting groove [16]. The groove diameter being smaller than the droplet’s diameter forces the droplet to squeeze and no longer adopt its relaxed shape. This deformation increases the interfacial area of the droplet and leads to an increase in its free energy [15]. In order to efficiently guide the droplets along the rails, the maximal force due to the surface energy gradient should be kept bigger than the hydrodynamic drag force. The rail starts from the sorting junction and ends at the beginning of the droplet size filtering area.

##### Droplet Size Filtering

In order to increase the trapping and pairing efficiency of our device, we added a passive filtering mechanism using the railing system already present (Figure 2ii). This size-based binary filtering enabled us to position the two different droplet types towards their respective traps. This filtering mechanism [7] is based on the Laplace trapping principle [22] defined by the Young-Laplace equation:(1)$$\Delta p=\frac{2\gamma }{R}$$
where *R* is the radius of curvature of the interface and *γ* the interfacial tension between the two phases.

When a droplet enters a narrow channel, due to the changing cross-sectional area of the groove, the radius of curvature of the interface increases and the Laplace pressure decreases, whereas the rest of the droplet will experience a higher Laplace pressure (indeed, entering the groove leads to a decrease in the total surface area of the droplet and a reduction in its surface energy). As a result, a net force is induced on the droplet pointing towards the energy minimum. In this passive filtering mechanism, two rails of different widths (the narrow rail on which droplets arrive is 10 ± 0.5 μm wide while the second rail onto which big droplets are deviated is 20 μm wide) are separated by a small gap (this 10 ± 0.5 μm gap defines the threshold for the droplet size filtering). All droplets arrive from the narrow rail. Big droplets, due to their larger radius, will come in contact with the wider rail and follow this other rail, as it will enable them to decrease their surface energy, while small droplets will never encounter the wider rail and remain on the narrow rail.

##### Droplet Trapping and Pairing

Once the droplets arrive at the trapping area (Figure 2iii) our goal is to trap pairs of droplets from different sizes using specifically designed floating trap arrays (FTA) [13,14,23]. FTAs are passive droplet trapping systems that rely on the buoyancy of droplets over the surrounding oil due to the lower density of the aqueous phase contained in the droplet. Efficient trapping is achieved by optimization of the geometric parameters of the trap, such as: trap diameter (*d_trap_* in Figure 2iii right), trap height (*h_trap_*) and channel height (*h_channel_*). Optimally, *h_trap_* should be sufficient enough (close to the droplet diameter *D*) to avoid clogging, but not too deep in order to trap only single droplets. In addition, a *h_trap_* parameter deeper than twice the droplet diameter will lead to inefficient droplet trapping by floatation under high flow rates (>7 μL/min). In order to achieve flow reduction and thus increase trapping efficiency, as well as enable the incorporation of electrodes (used for the merging step) in the microchannels, we decided to widen the channels in the trapping area to 364 μm. It was shown that ideal geometric parameters follow the rule: *d_trap_* = *h_trap_* = *h_channel_* = 1.17*D* [14]. The designed trapping system is also used for the pairing, and thus each trap consists of two traps used to pair one droplet of each size, *h_trap_* = *h_channel_* = 40 ± 0.3 μm for small droplets *d_trap_* (small) = 40 ± 0.5 μm, while for big droplets *d_trap_* (big) = 60 ± 0.5 μm (Figure 2iii right). According to the relation found previously, the ideal droplet size for the small traps should be 34.2 μm in diameter and 51.3 μm for the big droplets. The design used for the experiments presented below consists of a trapping area with 300 traps; this number of traps can be further increased up to thousands of traps.

FTAs have the advantage of relying on simple passive trapping and pairing mechanisms, not needing droplet synchronization and enabling a simple release of the trapped droplets compared to other droplet pairing mechanisms such as droplet clustering [24], trapping [25] or on flow droplet pairing [26]. The release of the droplet can be achieved by simply flipping the chip over and relying again on the buoyancy of the droplets over the oil [14], or by using a UV laser to generate cavitation in the target wells [27].

##### Droplet Merging

Once all traps are filled with droplet pairs, the droplets are merged in order to generate an array of single droplets containing two particles from different populations. Droplet coalescence (or droplet merging [28]) can either rely on passive mechanisms using the structure of the microchannel [29], the surface properties of droplets [30] or on active mechanisms (electric [31], thermal [32], magnetic [18]). We decided to use electrocoalescence (EC) as it has already been widely used and studied; additionally, it does not alter the content of the droplet during merging compared to chemical induced coalescence. As reported in the literature [33,34], droplet EC occurs when there is a difference in conductivity and/or permittivity between the dispersed and the continuous phase under the application of an electric field. The phenomenon arises due to polarization charges accumulated at the droplet interface, which can induce droplet–droplet interactions, leading to droplet deformation and coalescence. When two droplets are in close proximity and subjected to an electric field, the surfactant molecules (having dipolar head-groups) can be displaced or re-aligned along the field lines [35]. This can induce a destabilization of the interface that contacts the droplets, leading to the rapid fusion of the two emulsions. In the presented device, the electric field is applied between the two electrodes positioned on either side of the traps. The merging takes place over the whole trap array simultaneously. After merging, the array contains droplets co-encapsulating two different aqueous phases.

### 2.2. Design Droplet Device

The microfluidic device presented here consists of three inlets (two for oil and one for the aqueous phase) and two outlets (one for the waste and one for the droplets with double aqueous phase content). There are two pairs of electrodes used for the merging and the sorting of droplets.

### 2.3. Device Fabrication

The device is made out of polydimethylsiloxane (PDMS) microchannels closed with a glass slide patterned with titanium and platinum (Ti-Pt) electrodes (process flow available in the Appendix A). The fabrication process is as follows: Silicon (Si) masters are patterned with a soft photolithography step followed by a silicon etching step. The Si wafer is primed with hexamethyldisilane (HMDS), then 2 μm of AZ1512 HS (AZ 1500 series, MicroChemicals, Ulm, Germany) are spun and baked (ACS200 Gen3, Süss MicroTec, Garching, Germany). The wafers are then exposed to ultraviolet light through direct writing (VPG200 Photoresist Laser Writer, Heidelberg) to pattern the channels. After resist development, the Si is etched to a final height of 40 μm using the Bosch process (AMS 200SE, Adixen, EPFL, Switzerland). The photoresist is then stripped with 5 min Oxygen plasma at 500 W. The second layer of the master mold (for the rails and traps) consists of a 40 μm layer of SU-8 (MC3050, MicroChemicals) aligned over the previously patterned Si features for the microchannels (MA6Gen3, Süss MicroTec).

The final Si mold is silanized at room temperature overnight using Trichloro (1H,1H,2H,2H-perfluorooctyl) silane (PFOTS, Sigma-Aldrich, St. Louis, MO, USA). Devices are made by mixing the curing agent and PDMS polymer (PDMS 184 Sylgard, Dow Corning, CA, USA) at a ratio of 1:10 (wt/wt). The mixture of PDMS is then mixed and degassed in a vacuum chamber for 10 min, before being poured on the masters and degassed again prior to curing for at least 4 h at 80 °C.

After PDMS curing the masters are removed. The patterned PDMS slab is then cut out into individual chips, and inlets/outlets (both forming the microchannels and electrodes) are punched with a biopsy puncher of 0.75 mm in outer diameter.

The electrode patterned glass slides are made using a Borofloat wafer on which 20 nm of Ti and 200 nm of Pt were sputtered (SPIDER 600, Pfeiffer, EPFL, Switzerland). The metallized wafer is then patterned (photolithography, coating, exposure and development of AZ1512HS 2 μm) and etched using ion beam etching (IBE350, Veeco Nexus, EPFL, Switzerland) according to the final merging electrode design. The patterned glass wafer is then diced into individual glass slides (DAD dicing machine).

The PDMS channels and patterned glass slides are cleaned using frosted tape and bonded with oxygen plasma at 530 mTorr, 29 W for 45 s.

### 2.4. Microfluidic Chip Priming

Each microfluidic device was only used once and required appropriate priming in order to improve the hydrophobicity of the device before use. Aquapel priming solution was used to prime the devices. The solution was manually flushed and left in the channel for 10–30 s. The surface treatment solution was then removed by flushing air in the device. Finally, the device was rinsed with oil (QX200 Droplet oil generation for EvaGreen, BioRad, CA, USA).

### 2.5. Aqueous Phase Suspension

The aqueous phase used for the experiments is water-based food dye (red and blue).

### 2.6. Experimental Setup and Device Operation

The experiments were imaged with a Nikon Eclipse TE 300 (MicroscopyU, Tallahassee, FL, USA) inverted microscope using a 5X objective, and with a MQ003MG-CM camera (Ximea, Lakewood, CO, USA). The camera was controlled with Ximea Corp software acquiring at 500 frames per second (FPS). A Fluigent Flow EZ pressure controller ranging from 0 mbar to 1000 mbar was used to apply pressure on the glass vial containing the aqueous solution. Two Fluigent MFCS pressure controllers ranging from −800 mbar to 1000 mbar were used to apply pressure on the glass vials containing the oil (QX200 Droplet oil generation for EvaGreen, BioRad). 500 μm inner diameter Polytetrafluoroethylene (PTFE, IDEX 1569) tubing was used to connect the glass vials to the microfluidic chip through a custom-made vial cap. For simplification purposes:

*P*_aq_ = aqueous phase inlet pressure

*P*_oil_ = droplet generation oil inlet pressure

*P*_spacing_ = droplet spacing oil inlet pressure

## 3. Results

We decided to use the following pressure combinations for the two droplet sizes studied: *P*_aq_ = 55 mbar, *P*_oil_ = 113 mbar and *P*_spacing_ = 0 mbar (droplet generation rate of 10 droplets/sec) for small droplets, and *P*_aq_ = 65 mbar, *P*_oil_ = 60 mbar and *P*_spacing_ = 68 mbar (droplet generation rate of 60 droplets/sec) for big droplets. This corresponds to droplet volumes of 94 ± 2 pL and 104 ± 2 pL, respectively. The two droplet sizes will be hereafter identified according to their radius in the confined configuration (height confinement). Hence, a radius of 23.5 μm and 33.5 μm for the small and the big droplets, respectively.

### 3.1. Droplet Size-Based Filtering

We investigated the effect of the gap distance (*d*) between the two rails and the height of the rail (*h*) on the filtering efficiency and the size threshold at which droplets will be filtered.

Droplets are introduced in the filtering region from the narrow rail, then droplets are sorted between the different rails (narrow and wide) according to their size. Small droplets do not deform enough to encounter the wider rail and thus be deviated (Figure 3A(i)). However, big droplets are deformed by the wider rail and thus face the lower Laplace pressure, which enables the transfer of big droplets to the wider rail (Figure 3A(ii)). All droplets with radii larger than 35 μm are shifted to the wider rail (high-pass filtering), and droplets with radii smaller than 20 μm remain entirely on the narrow rail (low-pass filtering), as shown in Figure 3B.

Droplet filtering analogous to the high or low-pass filter of an electric circuit was investigated by varying the distance between rails (*d*) and the rail height (*h*). Three distances between the upper and lower rails were compared to study the effect of rail separation on filtering (Figure 3C) while keeping the height of the rail constant (*h* = 40 μm). By increasing the distance between the rails, the required droplet deformation becomes larger, thus shifting the threshold radius of droplet deviated to the lower rail [7]. Indeed, for rails with a distance of 0 μm, 100% of droplets with radii larger than 17 μm are deviated to the lower rail. This threshold radius increases to 33 μm for rails spaced by 10 μm, and for rails spaced by 20 μm the droplet radius for which all droplets are deviated to the wider rail was never reached, as droplets big enough could never be generated.

The effect of the rail height (*h*) on the droplet radii filtering threshold was also investigated (Figure 3D). Two different rail heights (40 μm and 60 μm) were studied with the same spacing of *d* = 10 μm; for both conditions, the droplet radii threshold at which all droplets are deviated to the wider rail is approximately 33 μm. However, the threshold at which small droplets are deviated will be shifted. For 40 μm high rails, droplets with radii smaller than 24 μm will remain on the upper rail, while for 60 μm high rails this threshold radius is shifted to 27 μm. The effect of the rail height is interesting as it enables us to define the steepness of the filter.

For our final application, the combination of the rail spacing of 10 μm and rail height of 40 μm was retained, as it enables us to have a filtering threshold in the middle of our droplet size generation range and to discriminate our small droplets of radius 23.5 μm from the big droplets of radius 33.5 μm.

### 3.2. Droplet Pairing

To optimize the droplet trapping efficiency (defined as the number of traps containing a droplet over the total number of traps available) we investigated the effect of the trap diameter (*d_trap_* (small) and *d_trap_* (big)) and the trap height (*h_trap_*) as well as the presence of rails.

The droplet sizes selected at the droplet generation step led to droplets of diameter 47 μm in the confined droplet (height confinement of droplets) configuration for small droplets (volume of 94 pL) and droplets of diameter 67 μm for the big droplets (volume of 104 pL). During our investigations, it appeared that the rails are crucial to improving trapping efficiencies for both droplet sizes (Figure 4E). Indeed, for identical trap dimensions (*h_trap_* = 40 μm, *d_trap_* (small) = 40 μm and *d_trap_* (big) = 60 μm) when no rail is present, droplets are spread over the whole trapping channel width, while the presence of single rails concentrates the droplets along specific flow lines in the trapping channel. The presence of two rails enables a filtering of the droplets according to size [7], and thus a precise positioning of the two droplet populations at different locations in the channel (Figure 4D). The trapping of the big droplets was optimized first and required fine tuning, as this step is performed first (Figure 4A,B) and the small droplet trapping is performed only when all big droplet traps are filled (Figure 4C). This step requires *d_trap_* (big) to be wide enough to attract big droplets from the rail towards the trap (Figure 4B) but also narrow enough to prevent double droplet trapping (big droplets go both in the small and big traps) [7,9]. It appears that the big droplet trapping efficiency in big traps is not influenced by the absence or presence of rails and remains at 100% in all conditions (Figure 4E). However, the rail presence enables us to trap all big droplets passing in the channel and thus prevent any droplet loss. Additionally, the double trapping (big droplets trapped in small traps) rate is drastically reduced from 70% to less than 20% when adding two rails (Figure 4E). A decrease in the double trapping of big droplets is of prime importance, as it prevents sample loss. The major advantage of the rail addition is for small droplet trapping; indeed, the efficiency of small droplet trapping increases from 0 to more than 90% when two rails are added. The rails play an important role, as they slow down the small droplets (thus decreasing the drag force over the trapping force) and also enable a good positioning of the droplets close to their respective trap.

We then investigated the effect of the trap’s diameter size on droplet trapping efficiency. We varied the big trap’s diameter from 40 to 70 μm, and in all cases the trapping efficiency of the big droplets was 100% (data not shown here). The small droplet trap diameter was varied between 40 and 50 μm while the big droplet traps had a diameter of 60 μm (Figure 4F). The optimal small droplet trap diameter is 40 μm, as it leads to a small droplet trapping efficiency of 100%.

During our experiments, it also appeared that the height of the trap plays a critical role in facilitating trapping (when a droplet is fully confined in the trap it undergoes less drag force and thus the trapping is more stable) but can also lead to double trapping.

The optimal combination of parameters maintained for the final design is *d_trap_* (small) = 40 μm, *d_trap_* (big) = 60 μm and *h_trap_* = 40 μm. Such parameters enable a 100% trapping efficiency for small and big droplets (Figure 4E) while limiting the double trapping to less than 5%. All flowing droplets on the rail are sequentially trapped starting at the first one until all traps are filled.

### 3.3. Droplet Merging

Once all traps are filled with pairs of droplets of different sizes, a square signal (AC) of 400 V (peak-to-peak) and 1 kHz frequency is applied between the chip electrodes. The droplet merging is instantaneous and has an efficiency of 100% (number of traps containing merged droplets over the total number of traps with droplet pairs). A video of the array electrocoalescence is available in the Appendix A.

## 4. Conclusions

We have successfully developed a microfluidic chip that can reliably and robustly:generate droplets of different sizes,pair droplets of different size in traps andmerge droplet pairs confined in traps.

Thanks to the combination of all those microfluidic elements on the device, we can generate a library of droplets containing two different aqueous phases. In terms of throughput, 25 min are required to generate such a library of 300 droplets (11 min are required per droplet type for droplet generation, filtering and trapping).

Full implementation of the steps on the same device was successfully demonstrated with two colored aqueous solutions.

The designed microfluidic device takes advantage of the already extensive knowledge and toolbox in droplet microfluidics. This novel design aims at efficiently, and with minimal sample loss, co-encapsulating two aqueous phases in the same droplet. This device could also be used with aqueous phases containing cells or particles. However, no biological proof of concept was performed in this study. Furthermore, an additional droplet sorting would be required in order to remove all the empty droplets and keep only the particle containing droplets.

The microfluidic device presented here improves the state of droplet pairing by combining FTA and railing systems in order to passively pair droplets from different compositions on a single chip.

## Figures and Tables

**Figure 1 micromachines-12-01076-f001:**
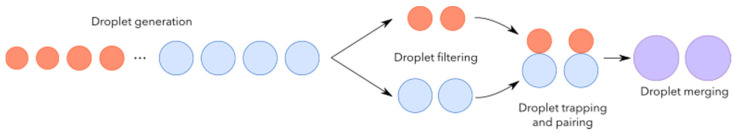
Droplet chip principle. The two aqueous phases are sequentially individually encapsulated in droplets from different sizes. The droplets are then filtered by size to position each droplet type towards its trapping location. The droplets are then trapped in pairs, each one containing a single droplet from each type. The droplet pairs are then merged.

**Figure 2 micromachines-12-01076-f002:**
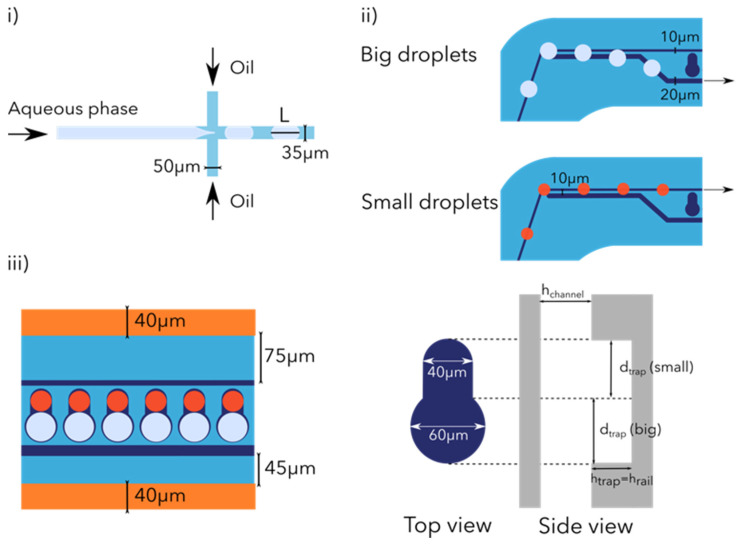
Schematic of some steps performed in the device. (**i**) Droplet generation. Scheme of the 2D 90° droplet generation system with the aqueous inlet and the oil inlets. The width of the microchannels is indicated and the droplet length in the confined configuration is indicated as L. (**ii**) Droplet railing. Scheme of the droplet size filtering system. Top: time lapse of a big droplet shifting from the original rail to the wider rail. Bottom: time lapse of a small droplet remaining on the original narrow rail. (**iii**) Droplet trapping and pairing. The left scheme shows the droplet trapping and pairing area with 5 traps filled with 2 droplets of each type. The electrodes used for droplet merging are represented (in orange). The right scheme shows top and side views of a single droplet trap; the critical trap dimensions for an efficient droplet trapping are indicated in the scheme. The trap height and the rail height are identical.

**Figure 3 micromachines-12-01076-f003:**
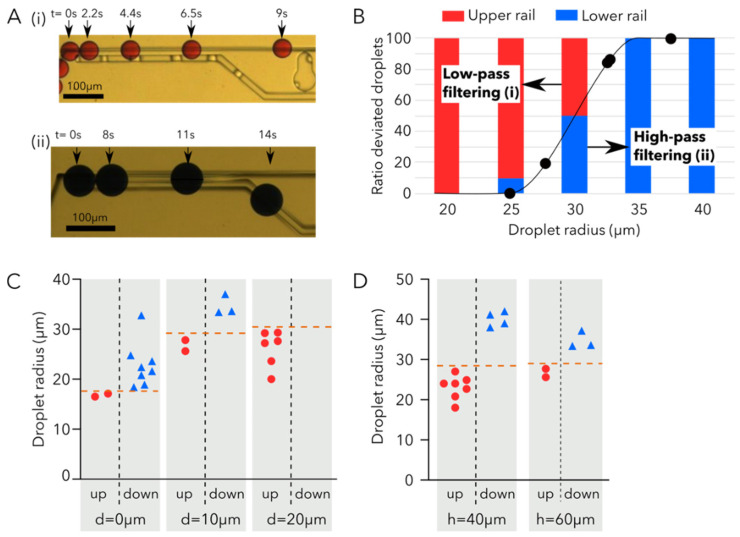
Droplet rail filtering system. (**A**) (**i**) Time-lapse of a small droplet continuing on the narrow rail. (**ii**) Time-lapse of a big droplet being deviated to the wider rail (**B**) Graph of the ratio of deviated droplets as a function of the droplet radius for the experimental case presented in (**A**). The droplets either remain on the narrow rail (low-pass filtering) or are deviated to the wider rail (high pass filtering) according to their size. (**C**) Number of droplets sorted according to the droplet radius for different gap distances between rails (called d). The droplet threshold transfer is indicated in orange. (**D**) Number of droplets sorted according to the droplet radius for different rail heights. The droplet threshold transfer is indicated in orange. Videos are available in Appendix A.

**Figure 4 micromachines-12-01076-f004:**
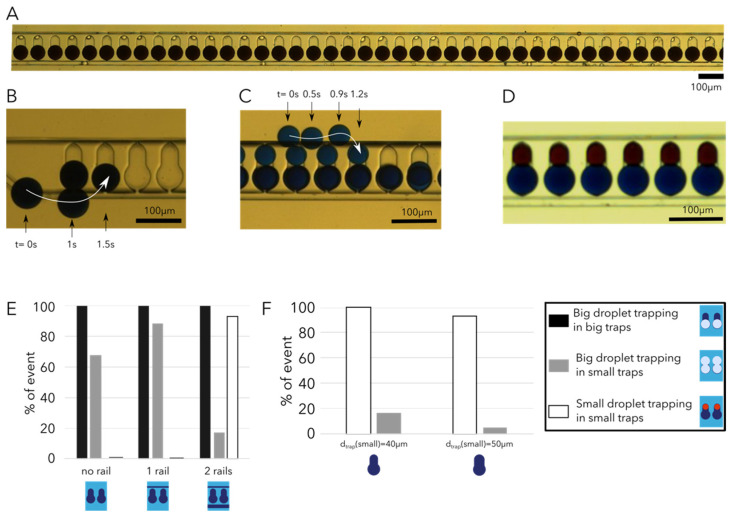
Droplet trapping array. (**A**) Brightfield image of big droplets trapped. (**B**) Time-lapse of a big droplet being trapped in the second trap of the array. (**C**) Time-lapse of a small droplet being trapped at the 90th trap of the array. (**D**) Brightfield image of six traps containing a single droplet of each type (two different dyes). The white arrows indicate the trajectories of the droplets. (**E**) Histogram of the big droplet trapping efficiency in big traps and small traps (double trapping) as well as the small droplet trapping efficiency in small traps under the conditions of no rail, a single rail or two rails present, for the same trap dimensions (big trap diameter of 60 μm, small trap diameter of 40 μm and trap height of 40 μm). (**F**) Histogram of the small and big droplet (double trapping) trapping in small traps according to the small droplet trap diameter (big traps have a diameter of 60 μm and height of the traps 40 μm). Videos are available in the Appendix A.

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
