# Peer review of "Microfluidic Device for Droplet Pairing by Combining Droplet Railing and Floating Trap Arrays"

_micromachines, 2021, doi:10.3390/mi12091076_

Round 1

Reviewer 1 Report

In this manuscript, authors introduced a microfluidic device for droplet size based filtration as well as paring and merging by combining droplet railing and floating trap arrays. 

Overall, this is a solid manuscript that proposes a new way of performing droplet sorting and pairing/merging steps in microfluidic droplets.  The capability described in the manuscript is important and will be valuable to the field of droplet microfluidics.  The paper is well written, figures are quite good.  The manuscript may be published after minor revision. In addition, the validation of maximum throughput or throughput vs efficiency of filtration and paring should be discussed in the result section.

Author Response

In this manuscript, authors introduced a microfluidic device for droplet size based filtration as well as paring and merging by combining droplet railing and floating trap arrays. 

Overall, this is a solid manuscript that proposes a new way of performing droplet sorting and pairing/merging steps in microfluidic droplets.  The capability described in the manuscript is important and will be valuable to the field of droplet microfluidics.  The paper is well written, figures are quite good.  The manuscript may be published after minor revision.

The authors thank the reviewer for his feedback.

In addition, the validation of maximum throughput or throughput vs efficiency of filtration and paring should be discussed in the result section.

The authors are grateful for the advices given by the reviewer and have added a sentence in the discussion

Reviewer 2 Report

Dear authors,

Please consider the suggested comments to improve the quality of the current version of the manuscript:

  1. The abstract needs to be incorporated with the gist of the complete work in the manuscript. In the current version of the abstract, only the summary of the work is mentioned in a very broad view. Instead, please include the specific details of the research study presented in the manuscript.
  2. In the abstract, please include the details of the limitations of the previous research studies/technologies made in the proposed research model. Also, specify the scientific advancements made in the current research to overcome those limitations.
  3. The abstract of a scientific research paper should be precisely mentioning the specific research question that is answered, experimental conditions, operational parameters, and results. Please explain and provide the specifications of the scientific questions answered in the proposed model.
  4. In the current version of the abstract, the scientific conclusions made based on the experimental results were not clearly mentioned. Also, the details of the experimental tests performed on the single droplets each containing a different aqueous phase that are paired and the corresponding efficiency, need to be clearly stated.
  5. In the introduction, please include more details of the peer studies performed on the ‘droplet railing’, to guide the reader to understand the importance of the research study performed. Also, include corresponding references in the text when mentioning the details.
  6. In the introduction, please include the knowledge gaps existing between the current research work and prior studies performed in the field. Very importantly, please specify the need for the current work presented in the manuscript.
  7. In the last paragraph of the introduction, kindly include the details of the broader impacts on the study made and the results achieved. It is very important to provide the future scope of the research performed to make a strong impact on the readers on the research performed/Study proposed.
  8. In figures 3A, 4A, 4B, 4C, & 4D, please include the scale to the images so that the readers can have the exact details mentioned in the images.
  9. In section-3: Results, kindly incorporate the appropriate logical reasoning and scientific conclusions made for the plots in fig-4E & 4F, also please use the ongoing research results of peers with appropriate references to support your arguments and statements.
  10. I think, section-4, is supposed to be named ‘Conclusions’. Please correct it.
  11. Please revise the manuscript with English grammar. There are few places that the manuscript needs to be improved with respect to English writing.

Author Response

The abstract needs to be incorporated with the gist of the complete work in the manuscript. In the current version of the abstract, only the summary of the work is mentioned in a very broad view. Instead, please include the specific details of the research study presented in the manuscript.

In the abstract, please include the details of the limitations of the previous research studies/technologies made in the proposed research model. Also, specify the scientific advancements made in the current research to overcome those limitations.

The abstract of a scientific research paper should be precisely mentioning the specific research question that is answered, experimental conditions, operational parameters, and results. Please explain and provide the specifications of the scientific questions answered in the proposed model.

In the current version of the abstract, the scientific conclusions made based on the experimental results were not clearly mentioned. Also, the details of the experimental tests performed on the single droplets each containing a different aqueous phase that are paired and the corresponding efficiency, need to be clearly stated.

We thank the reviewer for his comments on the content of the abstract. We have thus completely remodeled our abstract according to the advices given above.

In the introduction, please include more details of the peer studies performed on the ‘droplet railing’, to guide the reader to understand the importance of the research study performed. Also, include corresponding references in the text when mentioning the details.

In the introduction, please include the knowledge gaps existing between the current research work and prior studies performed in the field. Very importantly, please specify the need for the current work presented in the manuscript.

In the last paragraph of the introduction, kindly include the details of the broader impacts on the study made and the results achieved. It is very important to provide the future scope of the research performed to make a strong impact on the readers on the research performed/Study proposed.

We thank the reviewer for his comments on the introduction. We have accordingly modified the introduction to provide a stronger manuscript.

In figures 3A, 4A, 4B, 4C, & 4D, please include the scale to the images so that the readers can have the exact details mentioned in the images.

The scale bars have been directly added in the figures.

In section-3: Results, kindly incorporate the appropriate logical reasoning and scientific conclusions made for the plots in fig-4E & 4F, also please use the ongoing research results of peers with appropriate references to support your arguments and statements.

We thank the reviewer for this comment. We have further developed our reasoning on the mentioned figures and have added references to peer-revewied papers.

I think, section-4, is supposed to be named ‘Conclusions’. Please correct it.

We have modified the title of the "Discussion" section to "Conclusions"

Please revise the manuscript with English grammar. There are few places that the manuscript needs to be improved with respect to English writing.

We thank the reviewer for this comment and have corrected grammar and spelling mistakes.

Round 2

Reviewer 2 Report

Dear authors,
Thank you for updating the manuscript with recommended changes.